# Antibiotic residues and antibiotic-resistant bacteria detected in milk marketed for human consumption in Kibera, Nairobi

**Kelsey Brown[1], Maina Mugoh[2], Douglas R. Call[1], Sylvia Omulo[1,2]***

**1** Paul G. Allen School for Global Animal Health, Washington State University, Pullman, WA, United States of America, **2** Washington State University Global Health Kenya, Nairobi, Kenya

* Sylvia.omulo@wsu.edu

## Abstract

The use of veterinary antibiotics is largely unregulated in low-income countries. Consequently, food producers rarely observe drug withdrawal periods, contributing to drug residues in food products. Drug residues in milk can cause immunogenic reactions in people, and selectively favor antibiotic-resistant bacteria in unpasteurized products. We quantified the prevalence of antibiotic residues in pasteurized and unpasteurized milk, and antibiotic-resistant bacteria from unpasteurized milk sold within Kibera, an informal settlement in Nairobi, Kenya. Ninety-five milk samples (74 pasteurized and 21 unpasteurized) were collected from shops, street vendors or vending machines, and tested for the presence of β-lactam and tetracycline residues using IDEXX SNAP kits. MacConkey agar without- and with antibiotics (ampicillin, 32 μg/ml; tetracycline, 16 μg/ml) was used to enumerate presumptive *E. coli* based on colony morphology (colony forming units per ml, CFU/ml). β-lactam and tetracycline residues were found in 7.4% and 3.2% of all milk samples, respectively. Residues were more likely to be present in unpasteurized milk samples (5/21, 23.8%) compared to pasteurized samples (5/75, 6.8%); $P = 0.039$. Two thirds of unpasteurized samples (14/21, 66.7%) contained detectable numbers of presumptive *E. coli* (mean 3.5 $Log_{10}$ CFU/ml) and of these, 92.8% (13/14) were positive for ampicillin- (mean 3.2 $Log_{10}$ CFU/ml) and 50% (7/14) for tetracycline-resistant *E. coli* (mean 3.1 $Log_{10}$ CFU/ml). We found no relationship between the presence of antibiotic residues and the presence of antibiotic-resistant *E. coli* in unpasteurized milk sold within Kibera ($P > 0.2$).

**Data Availability Statement:** All relevant data are within the manuscript and its Supporting Information files.

## Introduction

Antibiotics can be used as "insurance" against livestock losses to disease, challenging the control of antibiotic use and antibiotic residues in food products. This situation is common in many low-income countries where the burden of infectious diseases drives the demand for antibiotics. In these settings, informal food markets are supplied with animals or animal products produced under limited antibiotic regulations, lack of enforcement of drug withdrawal periods, and absence of residue testing programs. For milk, depending on the drug

**Funding:** This study was supported by a summer research fellowship (grant #: N/A) from Washington State University's College of Veterinary Medicine. Kelsey Brown (student) was the recipient. The funders had no role in study design, data collection and analysis, decision to publish, or preparation of the manuscript.

**Competing interests:** The authors have declared that no competing interests exist.

formulation, the recommended withdrawal periods for ampicillin and oxytetracycline are 2 and 4 days [1], respectively. Adherence to these recommendations can be very expensive for persons living at the economic margins.

The presence of antibiotic residues in household and commercially available milk has been reported in East Africa [2–8]. β-lactams and oxytetracyclines, which are commonly used to treat mastitis and livestock respiratory diseases in this region, can trigger hyper-allergenic reactions in people if their residue concentration in consumed milk is sufficient [9–11]; maximum residue limits for amoxicillin and oxytetracycline are 4 ppb and 100 ppb, respectively [12]. Furthermore, for milk that is contaminated with pathogenic bacteria, antibiotic residues can favor the growth of antibiotic-resistant strains that may be directly ingested by the consumer. This is in addition to the risk posed when contaminated milk is exposed to temperatures that are optimal for bacterial growth (37–42˚C) [13].

In densely populated urban settlements, poor environmental hygiene and improper milk storage can contribute to milk contamination and proliferation of bacteria within milk, respectively. We estimated the prevalence of antibiotic residues in milk sold in Kibera, an informal settlement located within Nairobi. Kibera is serviced by a formal market which supplies pasteurized milk in sealed plastic bags or through automated vending machines, and by an informal market (small-scale farms) which supplies unpasteurized milk [14]. Given that most households in Kibera have no means to refrigerate milk, they are likely to encounter conditions that are ideal for growth of high-density populations of antibiotic-resistant bacteria in stored milk. To assess the degree to which this problem may arise in communities like Kibera, we collected pasteurized and unpasteurized milk samples from local vendors and tested them for antibiotic residues and bacterial counts (colony forming units per ml; CFU/ml). Bacterial counts were log transformed (base 10).

## Materials and methods

### Sampling

During September 2015, milk samples were purchased from formal and informal vendors serving Soweto and Gatwekera villages in Kibera. Milk samples were purchased from vendors trading within a 200 m radius from households that were participating in a longitudinal study on antimicrobial resistance. Sample collection occurred over a 2-week period, primarily between 9 and 11 a.m. Once collected, samples were transported on ice to a microbiology laboratory located in Kibera within two hours of collection.

### Residue testing

All samples were transferred into sterile 50-ml conical tubes and tested for the presence of β-lactam and tetracycline residues by using IDEXX SNAP kits (IDEXX Laboratories Inc., Maine, USA) following manufacturer instructions [15,16]. These commercial test kits provide rapid presence/absence results at a sensitivity approaching 50 ppb and cross-react with a variety of β-lactam and tetracycline analogues, respectively [17]. Residue testing was completed on the day of sample collection and results were recorded as "positive" or "negative" for presence of the respective antibiotic residue. Milk spiked with tetracycline and ampicillin at 20 μg/ml (20 ppm) was used as the positive control for the SNAP kits.

### Bacteriology

The total number of presumptive *E. coli* and antibiotic-resistant *E. coli* was also determined for each sample on the day of sample collection. Unpasteurized milk samples were serially diluted

(10-fold) with phosphate-buffered saline and 50 µl of the $10^0$ to $10^{-3}$ dilution was plated onto MacConkey agar plates with no antibiotic, with ampicillin (32 µg/ml) and with tetracycline (16 µg/ml). The latter two plates selected for ampicillin-resistant (Amp[R]) or tetracycline-resistant (Tet[R]) *E. coli*, respectively. Plates were incubated at 37˚C for 18–24 hours and presumptive *E. coli* identified by colony morphology [18]. Plates with 10–100 colonies were selected for colony counts, and the colony-forming units (CFU) per mL recorded for each sample. When fewer than 10 colonies were observed at the $10^0$ dilution, all visible colonies were counted. If colony density greatly exceeded 100 colonies at the $10^{-3}$ dilution, the refrigerated left-over sample was diluted further and re-plated. Prior to sample collection, five pasteurized, packaged milk samples were purchased and plated as described, and no bacteria were detected. Consequently, no additional pasteurized milk samples were tested for bacterial growth.

## Minimum detection sensitivity

To determine the analytic detection sensitivity of the methods employed in this study, we serially diluted (10-fold) 2.5 x $10^9$ CFU of *E. coli* with whole pasteurized milk. Four dilutions ($10^0$ to $10^{-3}$) were plated onto MacConkey agar using the spread plating technique and incubated at 37˚C for 18 hours. The minimum number of detectable *E. coli* was determined from the plate containing the highest milk dilution with visible colonies.

## Data analysis

A Wilcoxon rank-sum test was used to compare the number of unpasteurized samples relative to the presence of antibiotic residues and antibiotic-resistant *E. coli*. To compare the correlation between counts (CFU/ml) of Amp[R] and Tet[R] *E. coli*, zero counts were transformed to a random number between 0 and 650 (uniform distribution) to account for detection sensitivity limits, and all values were log-transformed (base 10) before the analysis. Statistics were calculated by using Stata software (ver. 15.1, StatCorp LLC, College Station, TX).

## Results

In total, 96 milk samples were collected, 75 of which were purchased from shops (pasteurized) and 21 from mobile vendors (unpasteurized). Pasteurized samples were mainly sold in 250–500 mL sealed plastic packages, while unpasteurized samples were measured in a 250 mL glass and transferred into thin plastic bags (Fig 1). One pasteurized milk sample was excluded from the analysis due to fermentation. Ten of the total 95 milk samples (10.5%) tested positive for antibiotic residues, including seven (7.4%) which were positive for β-lactam residues and three (3.2%) for tetracycline residues; none were positive for both. Residues were more likely to be present in unpasteurized samples (5/21, 23.8%) compared with pasteurized samples (5/74, 6.8%); $P = 0.039$. Among the 21 unpasteurized samples, 14 samples (66.7%) contained detectable numbers of presumptive *E. coli* colonies (mean 3.5 $Log_{10}$ CFU/ml) out of which 92.8% (13/14) and 50% (7/14) were positive for Amp[R]–(mean 3.2 $Log_{10}$ CFU/ml) and Tet[R]–*E. coli* (mean 3.1 $Log_{10}$ CFU/ml), respectively. No *E. coli* were recovered from seven of the unpasteurized samples (S1 and S2 Tables).

The minimum detection sensitivity of our methods was 2.8 $Log_{10}$ CFU/ml. Unpasteurized milk samples had *E. coli* counts ranging from 1.1–7.5 $Log_{10}$ CFU/ml, while the counts of Amp[R] *E. coli* or Tet[R] *E. coli* ranged from 1.3–6.9 and 2.0–6.7 $Log_{10}$ CFU/ml, respectively (Fig 2). There was a significant correlation between the number of Amp[R] and Tet[R] *E. coli* ($r^2 = 0.81$, $P = 0.001$). The presence of antibiotic residues was not associated with the number of antibiotic-resistant *E. coli* ($P > 0.5$ for all comparisons).

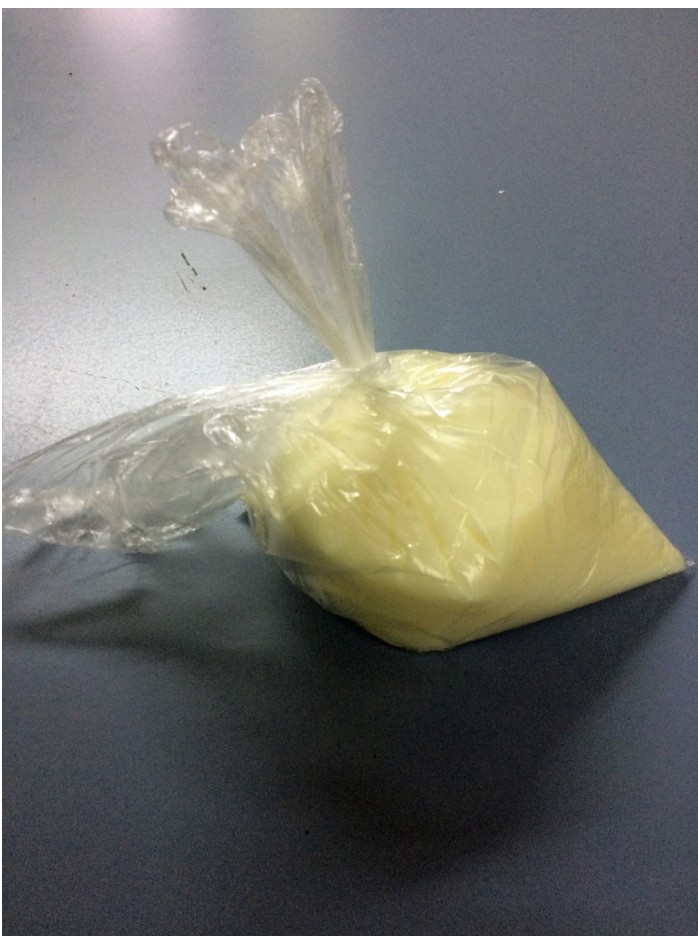
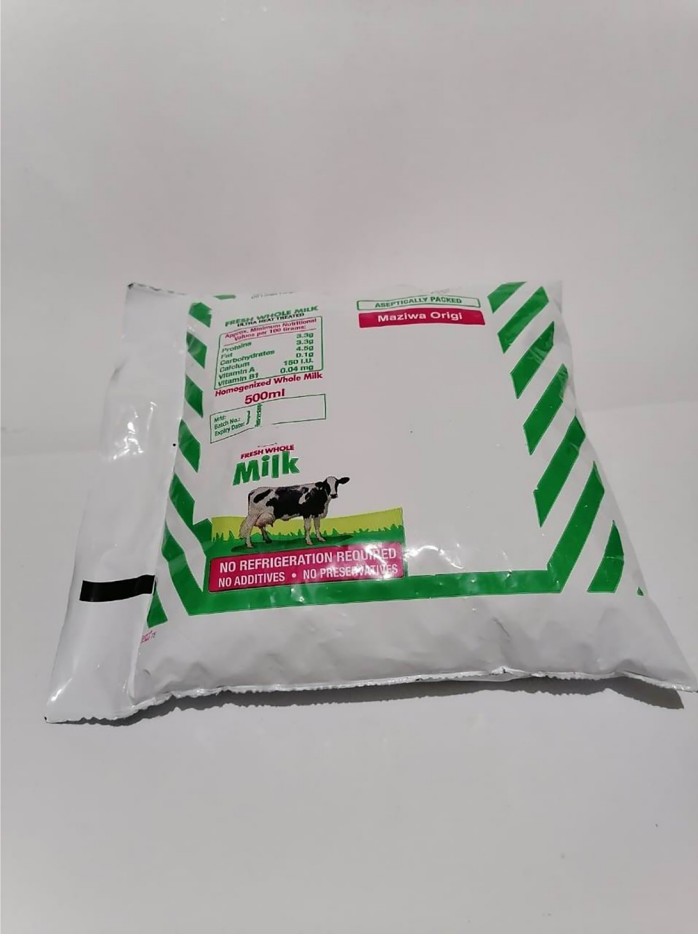

**Fig 1.** Examples of milk samples tested; (a) unpackaged/unpasteurized milk and (b) packaged/pasteurized milk (modified version of original packaging; used for illustrative purposes only).

## Discussion

Unpasteurized milk has a potential role in disseminating both pathogens and antibiotic-resistant bacteria to people through several mechanisms. First, antibiotic-resistant bacteria can be directly acquired through ingestion of milk contaminated with these bacteria [19]. In this study, 67% of unpasteurized milk samples contained *E. coli*, most of which were resistant to ampicillin and/or tetracycline. Further, the strong correlation between the number of $Amp^R$ and $Tet^R$ *E. coli* suggests that these were likely multi-drug resistant strains. Consuming just one cup of milk contaminated with $10^6$ antibiotic-resistant bacteria per ml can result in inoculation with over $10^8$ bacteria, a problem that can be prevented through pasteurization. Nevertheless, where storage is poor post-pasteurization hygiene problems (e.g., use of contaminated containers) can lead to re-contamination. Beyond transmission of antibiotic-resistant bacteria, livestock serve as reservoirs for multiple gastrointestinal pathogens of public health concern.[22] Ingestion of these pathogens in unpasteurized milk can increase antibiotic use by the consumer, contributing to the emergence of AMR [20–22].

Antibiotic residues in milk can select for antibiotic-resistant bacteria within milk itself, which can then be transmitted directly to people through ingestion. It was unclear from this study if this mechanism is important since we found no correlation between the presence of

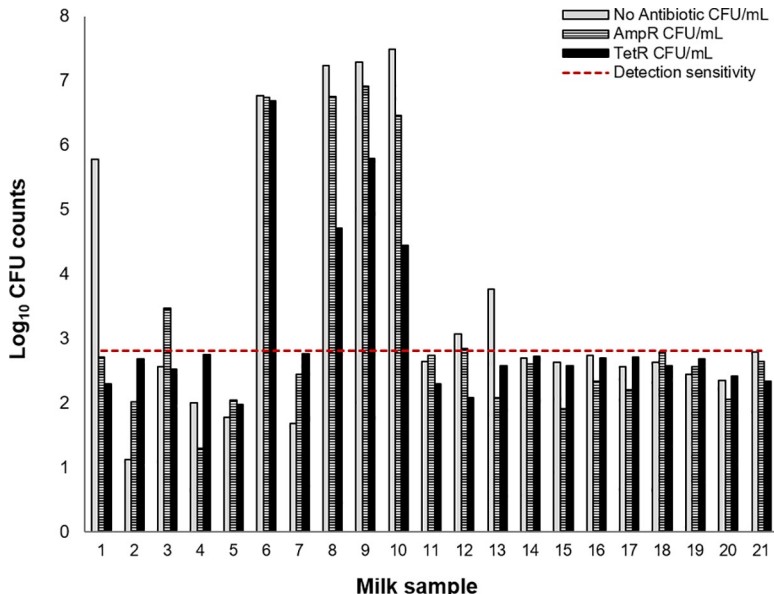

**Fig 2. Total, Amp^R and Tet^R *E. coli* counts (Log$_{10}$ CFU/mL) for individual unpasteurized milk samples (n = 21).** No bacterial growth was observed across the three media types for samples 2, 7, 17–21. An explanation for the variation shown below the detection limit is provided under the methods section.

antibiotic residues and that of antibiotic-resistant *E. coli*. This may be a limitation of the small sample size considered in this study. Additionally, being that the samples collected in this study were obtained from vendors rather than individual households, they unlikely represent the full range of storage conditions that may exist within Kibera. Fortunately, most of the milk samples (n = 75) collected in this study were pasteurized, perhaps reflecting a higher prevalence of vendors selling pasteurized than unpasteurized milk in Kibera, and a relative affordability of packaged pasteurized milk. Further, the relationship between antibiotic residues and antibiotic-resistant bacteria is dose-dependent [23]. The SNAP tests used in this study simply allowed a "positive" or "negative" classification of samples without quantifying the concentration of residues within milk samples classified as "positive". Additional work is needed to quantify antibiotic residues within milk to re-examine their relationship with antibiotic-resistant bacteria.

The consumption of antibiotic residues in milk can potentially select for antibiotic-resistant bacteria within a consumer's gut microflora, a mechanism that has yet to be fully investigated [24]. A study that administered a therapeutic dose (10 mg/kg) of oxytetracycline intramuscularly to groups of cows reported antibiotic residue concentrations in milk as high as 1.92 μg/ml. This concentration falls within a range that can selectively favor antibiotic-resistant *E. coli* [25] and is likely sufficient to do so after ingestion of contaminated milk depending on rates of absorption and dissipation [26]. In this study, the prevalence of β-lactam (7%) and tetracycline (3%) positive samples was 9 and 4 times higher than the prevalence of residues (0.8%) reported in U.S. milk in 2012 [27]. Residues were observed in both pasteurized and unpasteurized samples, indicating that residue control needs to be focused on all producers, although there is a clear trend towards a lower prevalence of contamination for pasteurized products. In Kenya, boiling is commonly used when consumers prepare milk for consumption, but this practice does not appear to affect presence of residues in milk [28].

Aside from the potential antibiotic-resistance consequences of having antibiotic residues in milk, ingestion of these residues can cause allergic reactions, carcinogenicity, hepatotoxicity,

bone marrow toxicity, and reproductive disorders [9,11,28–31]. Limiting antibiotic residues in milk will require a multimodal approach including education of producers, stricter oversight of antibiotic sales and withdrawal times (in milk, ampicillin and oxytetracycline withdrawal times are 2 days and 4 days following injection, respectively [1]), stronger surveillance of residues and AMR in food animal products, and increased awareness and concern of AMR and its pathways of dissemination amongst policy makers and veterinary officials [3,32].

There were several limitations to this study. Firstly, the SNAP tests used to detect antibiotic residues (presence/absence) required a subjective interpretation of the results; we classified samples as negative unless the test was very clearly positive. Secondly, we cannot conclusively tell how the milk affects consumers, who are likely to process milk prior to consumption or consume it later after purchasing, given that milk was tested soon after its purchase from vendors. It is also possible that antibiotic residues may have a greater effect on microbial contaminants the longer the consumer stores milk. We purchased and tested milk directly from the vendor and did not consider consumer behaviors and practices. We also acknowledge that sample collection was opportunistic, rather than random, which could introduce bias to these findings.

## Supporting information

**S1 Table. Study data for all milk samples collected.**
(XLSX)

**S2 Table. Study data for unpasteurized milk samples only.**
(XLSX)

## Acknowledgments

The authors would like to thank the field team working within Kibera for collecting the milk samples for this project.

## Author Contributions

**Conceptualization:** Kelsey Brown, Douglas R. Call, Sylvia Omulo.

**Data curation:** Maina Mugoh, Sylvia Omulo.

**Formal analysis:** Douglas R. Call.

**Funding acquisition:** Kelsey Brown.

**Methodology:** Kelsey Brown, Maina Mugoh, Sylvia Omulo.

**Resources:** Maina Mugoh, Douglas R. Call.

**Supervision:** Douglas R. Call, Sylvia Omulo.

**Writing – original draft:** Kelsey Brown.

**Writing – review & editing:** Maina Mugoh, Douglas R. Call, Sylvia Omulo.

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
