## [Decision Letter · Decision Letter 0]

13 Feb 2020

PONE-D-19-29752

Antibiotic Residues and Antibiotic-Resistant Bacteria in Milk Marketed for Human Consumption in Kibera, Nairobi

PLOS ONE

Dear Dr. Omulo,

Thank you for submitting your manuscript to PLOS ONE. After careful consideration, we feel that it has merit but does not fully meet PLOS ONE’s publication criteria as it currently stands. Therefore, we invite you to submit a revised version of the manuscript that addresses the points raised during the review process.

We would appreciate receiving your revised manuscript by Mar 29 2020 11:59PM. To enhance the reproducibility of your results, we recommend that if applicable you deposit your laboratory protocols in protocols.io, where a protocol can be assigned its own identifier (DOI) such that it can be cited independently in the future. For instructions see: http://journals.plos.org/plosone/s/submission-guidelines#loc-laboratory-protocols

We look forward to receiving your revised manuscript.

Kind regards,

James E. Wells, PhD

Academic Editor

PLOS ONE

Journal Requirements:

1. We understand that you purchased milk from local markets for this study. In your Methods section, please provide additional details regarding the source of this material. Please provide the geographic coordinates and names of the purchase locations (e.g., shops, vendors), if available, as well as any further details about the purchased items (e.g., quantity, source origin) to ensure reproducibility of the analyses.

2. Thank you for including the following funding information within the acknowledgements section of your manuscript; "This work was funded by the Paul G. Allen School for Global Animal Health, and by the Washington State University College of Veterinary Medicine Summer Research Fellowship Program. "

3. 

We note that you have indicated that data from this study are available upon request. PLOS only allows data to be available upon request if there are legal or ethical restrictions on sharing data publicly. For more information on unacceptable data access restrictions, please see http://journals.plos.org/plosone/s/data-availability#loc-unacceptable-data-access-restrictions.

Additional Editor Comments (if provided):

The reviewers observe merit in the study, but there are significant concerns relative to the methods presented and the statistical analyses conducted. The authors need to clearly state how the study was conducted and the analyses that were done. Furthermore, the data presentation could be improved and statistical analyses need to be conducted properly as noted by Reviewer 1.

Reviewers' comments:

Reviewer's Responses to Questions

**Comments to the Author**

1. Is the manuscript technically sound, and do the data support the conclusions?

Reviewer #1: Yes

Reviewer #2: Yes

2. Has the statistical analysis been performed appropriately and rigorously? 

Reviewer #1: No

Reviewer #2: Yes

3. Have the authors made all data underlying the findings in their manuscript fully available?

Reviewer #1: No

Reviewer #2: No

4. Is the manuscript presented in an intelligible fashion and written in standard English?

Reviewer #1: Yes

Reviewer #2: Yes

5. Review Comments to the Author

Reviewer #1: The manuscript by Brown et al presents results of public health importance. They reported antibiotic residues in 10% of ready to consume milk, and majority of unpasteurized milk samples containing high bacterial load including antibiotic resistant bacteria. Although the study is of great interest, the methods are vaguely presented, results are inconsistent, poorly presented and discussed. Please see general and specific comments below.

Abstract

Line 32: please change “hypoallergenic” to “hyperallergenic” as used in the body of the manuscript (line 61). I suggest using the more general term (without the specific type) “allergic”.

Line 43-44 and elsewhere in the text: would you present the CFU/ml values in log10 scale?

Line 43-44 and elsewhere in the text: how these percentages (note that in the text it was mentioned 92% for ampicillin line 131) were calculated? Please clearly state this in the materials and methods. How is it possible to calculate resistant fraction obtained on media supplemented with antibiotics from total bacterial count obtained from plain media? This biases towards the numerator as we already select for the resistant population, simply the methods by which the two values were determined are different. For the stated proportions replica platting would be appropriate. See more comment on this below.

Line 45: How did you arrive at the conclusion of “no evidence”? Please show statistical analysis.

Introduction

Please add literature (perhaps between lines 61 and 62) on the maximum residue level (MRL) allowed in milk for the beta-lactam and oxytetracycline.

Line 71: replace” with” by “which”

Materials and methods

Line 89: Please add manufacturer’s information (company, city, State or Country)

Line 99-100: what dilution(s) were plated? This is important since you used direct plating, too numerous to count indicates high bacterial load; on the other hand, most samples can be BDL if contamination is low which should be expected in milk (and water) and normally filtration is also used for detection.

Line 113: please correct the degree sign as 37°C

Why your analytical sensitivity of 650 CFU/ml is high? How many colonies per plate was your cut-off to count? Did you count 1CFU/plate and consider such a sample as positive, or how did you deal with? Please clarify your methods.

Data analysis

Was the “random number” substitution a form of imputation? Please state. Also discuss if this method would introduce a bias, for not adding it to observations with enumerable values.

What log scale was used? If log10 state so.

Results

Line 129-130: please provide statistical significance for the difference.

Line 130-131: as commented above please clarify how the percentages were calculated.

“Among unpasteurized samples, 14 samples (67%) contained presumptive E. coli colonies out of which 92.8% and 50% were positive for AmpR– and TetR–E. coli, respectively.” This is inconsistent with what was presented in the abstract: Line 42-44: “One third of unpasteurized samples (8/21, 38%) contained detectable numbers of presumptive E. coli (mean 9.2 x 106 CFU/ml) and of these, 87% were positive for ampicillin- (3.7 x 106 CFU/ml) and 50% for tetracycline-resistant E. coli (1.4 x 106/ml).”

Line 130: what does the phrase “contained presumptive” indicate and how was it derived? Is it enumeration positive (i.e. above BDL)? Please clarify.

Please mention that E. coli (wild or resistant strains) were not observed on enumeration plate or are BDL.

Line 133-134: please also include the mean values (preferably log10) for each media type used.

Line 136: given you have only tested two antibiotic classes, I would replace “multidrug-resistance” with co-selection.

Line 137: change “residuals” to residues.

Fig: Please use log10 scale since most studies report that way. In the title please add “by media type” to avoid confusion with drug residues. For the milk samples with no plot observations (example #2, 7, 17-21), were the three types of bacteria (generic, Amp resistant and Tet resistant E. coli) absent across the three media types? Please state. What do ME and MD mean? Please state.

Discussion

Line 146: please insert “unpasteurized” before “milk” and elsewhere in the discussion. Pasteurization removes this issue.

Line 146-148: again, you did not test individual colonies of the total E. coli you obtained on plain media to give you proportion resistant, which would only be possible through replica platting. Rather you isolated the three bacterial strains independently. So, present all your results on sample basis throughout the manuscript.

Line 161: should read “represent”

Reviewer #2: This is a modest but well executed study that examines the presence of antibiotic residues in milk from vendors in Nairobi and seeks to determine if there are any correlations between presence of residues, and presence of resistant bacteria. Essentially looking to inform the question of whether the residues, if present, might enrich for antibiotic resistant bacteria in the milk. There are a number of experimental design limitations in the current study that impact the interpretation of the results. However the authors clearly acknowledge these, and provide informative discussion points so that readers who might not be familiar with the details of antibiotic resistance in this system will be able to understand the limitations. There is a need to better understand antibiotic resistance in African countries, and this study provides information that informs question related to food safety and antibiotic resistance in a culturally relevant setting. The data need to be made available in supporting information.

6. PLOS authors have the option to publish the peer review history of their article (what does this mean?). If published, this will include your full peer review and any attached files.

Reviewer #1: No

Reviewer #2: No

---

## [Author Response · Author response to Decision Letter 0]

1 Apr 2020

Comment 1 

We understand that you purchased milk from local markets for this study. In your Methods section, please provide additional details regarding the source of this material. 

• Please provide the geographic coordinates and names of the purchase locations (e.g., shops, vendors), if available.

>> We have provided a study description for this community, but we cannot supply the requested geographic coordinates. This is because such coordinates are not available for the mobile vendors who sell raw milk or even for the formal vendors that typically erect a shop that is later moved or abandoned within a matter of months. As such, reporting such coordinates would not help in the replication of this study

• Further details about the purchased items (e.g., quantity, source origin) to ensure reproducibility of the analyses.

>> These details have now been included. We have also included a figure (not Fig 1) to illustrate the packaging of these milk samples. We did not collect data regarding the source origin of the milk samples we purchased.

Comment 2

Thank you for including the following funding information within the acknowledgements section of your manuscript; "This work was funded by the Paul G. Allen School for Global Animal Health, and by the Washington State University College of Veterinary Medicine Summer Research Fellowship Program. "

• Please remove any funding-related text from the manuscript and let us know how you would like to update your Funding Statement. Currently, your Funding Statement reads as follows: "The funders had no role in study design, data collection and analysis, decision to publish, or preparation of the manuscript."

>> Funding information deleted from the acknowledgement section

Comment 3 

The data should be provided as part of the manuscript or its supporting information, or deposited to a public repository. For example, in addition to summary statistics, the data points behind means, medians and variance measures should be available. If there are restrictions on publicly sharing data—e.g. participant privacy or use of data from a third party—those must be specified.

>> The dataset has been provided as part of supporting information.

Comment 4

The reviewers observe merit in the study, but there are significant concerns relative to the methods presented and the statistical analyses conducted. 

• The authors need to clearly state how the study was conducted and the analyses that were done.

>> Specific questions about the methods and analyses have been clarified in the revision as indicated under each specific comment below.

• Furthermore, the data presentation could be improved, and statistical analyses need to be conducted properly as noted by Reviewer 1.

>> We believe that we have addressed the clarity issues that were noted by the reviewer. Please note that the reviewer mistakenly concluded that we calculated the proportion of resistant bacteria by making a ratio of the CFU count from agar plates without antibiotic and agar plates with antibiotic. All our comparisons are on a sample basis (presence or absence of detectable antibiotic-resistant bacteria) as the reviewer suggests should have been done. We report total counts from the plates, but not a proportion of the plate without antibiotic. It is unclear to us why this confusion may have occurred.

Comment 5

Reviewer #1: The manuscript by Brown et al presents results of public health importance. They reported antibiotic residues in 10% of ready to consume milk, and majority of unpasteurized milk samples containing high bacterial load including antibiotic resistant bacteria. 

• Although the study is of great interest, the methods are vaguely presented, results are inconsistent, poorly presented and discussed. Please see general and specific comments below.

• Abstract

o Line 32: please change “hypoallergenic” to “hyperallergenic” as used in the body of the manuscript (line 61). I suggest using the more general term (without the specific type) “allergic”. 

>> we have replaced this term with “immunogenic”

o Line 43-44 and elsewhere in the text: would you present the CFU/ml values in log10 scale?

>> This has now been addressed.

o Line 43-44 and elsewhere in the text: how these percentages (note that in the text it was mentioned 92% for ampicillin line 131) were calculated? Please clearly state this in the materials and methods.

>> The calculation of all percentages has been clarified by indicating the specific counts that form the numerators and denominators. 

o How is it possible to calculate resistant fraction obtained on media supplemented with antibiotics from total bacterial count obtained from plain media? This biases towards the numerator as we already select for the resistant population, simply the methods by which the two values were determined are different. For the stated proportions replica platting would be appropriate. See more comment on this below.

>>As above, we are not sure why the reviewer drew this conclusion about the methods. There is no text describing calculation of a resistant fraction in terms of bacterial numbers. All proportion values that we report are relative to the proportion of positive or negative samples, not bacterial counts. For bacterial counts, we calculated total bacterial counts based on the corresponding plates from which these counts were made.

o Line 45: How did you arrive at the conclusion of “no evidence”? Please show statistical analysis.

>> We have revised this statement to read: “We found no relationship between the presence of antibiotic residues and the abundance of antibiotic-resistant E. coli”.

• Introduction

o Please add literature (perhaps between lines 61 and 62) on the maximum residue level (MRL) allowed in milk for the beta-lactam and oxytetracycline.

>> This information has been added

o Line 71: replace” with” by “which”

>> Corrected. Thank you.

• Materials and methods

o Line 89: Please add manufacturer’s information (company, city, State or Country)

>> This information has been added

o Line 99-100: what dilution(s) were plated? This is important since you used direct plating, too numerous to count indicates high bacterial load; on the other hand, most samples can be BDL if contamination is low which should be expected in milk (and water) and normally filtration is also used for detection.

>> The dilutions that were plated have been included in the revision.

o Line 113: please correct the degree sign as 37°C

>> This has been corrected. Thank you.

o Why your analytical sensitivity of 650 CFU/ml is high?

>> We agree that this value is higher than what would be expected with, for example, water, but it is an empirically derived value. We surmise that the count seems high because bacteria clump with milk components and thus a single colony from a milk sample potentially represents more than one bacterium. For the same reason, filtration would not have remedied this issue. Please note that if such clumping occurred, we consider this effect to be similar across all samples

o How many colonies per plate was your cut-off to count?

>> We primarily chose the dilution that gave us at least 10 colonies per plate. Counts below this threshold were only included for undiluted/neat milk samples. These thresholds have been included in the methods.

o Did you count 1CFU/plate and consider such a sample as positive, or how did you deal with? Please clarify your methods.

>> There was only one instance where one colony was observed in plate containing an undiluted milk sample. This plate had 5 colonies in the Mac agar without antibiotics, 1 in the Amp+ plate and no colonies in the Tet+ plate.

• Data analysis

o Was the “random number” substitution a form of imputation? Please state.

>> This was based on a random number drawn from a uniform distribution. This has been added to the manuscript.

o Also discuss if this method would introduce a bias, for not adding it to observations with enumerable values.

>> We used this method to avoid issues that arise when transforming small numbers to a log scale (negative axis values) and because it acknowledges that we should not treat a negative value as truly negative given that there is a minimum analytic sensitivity of this assay. Not adding this value has no consequence to statistical comparisons that are made on a sample basis (positive vs. negative). Failure to add a similar randomly selected number between 0 and 650 would potentially bias the final counts to a lower value, but this is unlikely to be significant as the counts increase on a log scale. 

o What log scale was used? If log10 state so.

>> This has now been addressed.

• Results

o Line 129-130: please provide statistical significance for the difference.

>> This information has been added

o Line 130-131: as commented above please clarify how the percentages were calculated.

“Among unpasteurized samples, 14 samples (67%) contained presumptive E. coli colonies out of which 92.8% and 50% were positive for AmpR– and TetR–E. coli, respectively.” This is inconsistent with what was presented in the abstract: Line 42-44: “One third of unpasteurized samples (8/21, 38%) contained detectable numbers of presumptive E. coli (mean 9.2 x 106 CFU/ml) and of these, 87% were positive for ampicillin- (3.7 x 106 CFU/ml) and 50% for tetracycline-resistant E. coli (1.4 x 106/ml).”

>>This was an error that has now been corrected.

o Line 130: what does the phrase “contained presumptive” indicate and how was it derived? Is it enumeration positive (i.e. above BDL)? Please clarify.

>>Please recall that we identified E. coli based only on colony morphology. While this works the majority of the time (cite your JMM paper), it is not perfect. We use “presumptive” as a qualifier to acknowledge this uncertainty.

o Please mention that E. coli (wild or resistant strains) were not observed on enumeration plate or are BDL.

>> This has now been addressed.

o Line 133-134: please also include the mean values (preferably log10) for each media type used.

>> This information has now been added.

o Line 136: given you have only tested two antibiotic classes, I would replace “multidrug-resistance” with co-selection.

>> This has been amended to make it clear that there is a probability that these were multi-drug resistant strains, even though we did not test for other resistance types. Additionally, we have moved this bit of statement to the discussion section to make it clear that this is our interpretation of this finding.

o Line 137: change “residuals” to residues.

>> This has been corrected. Thank you

o Fig: Please use log10 scale since most studies report that way. 

>> This has now been addressed.

o In the title please add “by media type” to avoid confusion with drug residues. 

>> We have changed values to log scale. We changed the title by inserting the word “detected” to eliminate confusion (although we are not certain that we follow the reviewer’s meaning with respect to the original title).

o For the milk samples with no plot observations (example #2, 7, 17-21), were the three types of bacteria (generic, Amp resistant and Tet resistant E. coli) absent across the three media types? Please state. 

>> This information has been included in the figure caption.

o What do ME and MD mean? Please state.

>> This information has been included in the figure caption.

• Discussion

o Line 146: please insert “unpasteurized” before “milk” and elsewhere in the discussion. Pasteurization removes this issue.

>> This has been amended as suggested.

o Line 146-148: again, you did not test individual colonies of the total E. coli you obtained on plain media to give you proportion resistant, which would only be possible through replica platting. Rather you isolated the three bacterial strains independently. So, present all your results on sample basis throughout the manuscript.

>> we are uncertain why the reviewer is making this argument since our results are presented on a sample basis. We report total counts with and without antibiotic in the agar, but we did not calculate a proportion of resistant bacteria based on these values.

o Line 161: should read “represent”

>> This has been corrected. Thank you

---

## [Decision Letter · Decision Letter 1]

6 May 2020

Antibiotic residues and antibiotic-resistant bacteria detected in milk marketed for human consumption in Kibera, Nairobi

PONE-D-19-29752R1

Dear Dr. Omulo,

We are pleased to inform you that your manuscript has been judged scientifically suitable for publication and will be formally accepted for publication once it complies with all outstanding technical requirements.

With kind regards,

James E. Wells, PhD

Academic Editor

PLOS ONE

Additional Editor Comments (optional):

Reviewers' comments:

Reviewer's Responses to Questions

**Comments to the Author**

1. If the authors have adequately addressed your comments raised in a previous round of review and you feel that this manuscript is now acceptable for publication, you may indicate that here to bypass the “Comments to the Author” section, enter your conflict of interest statement in the “Confidential to Editor” section, and submit your "Accept" recommendation.

Reviewer #1: All comments have been addressed

Reviewer #2: All comments have been addressed

2. Is the manuscript technically sound, and do the data support the conclusions?

Reviewer #1: (No Response)

Reviewer #2: Yes

3. Has the statistical analysis been performed appropriately and rigorously? 

Reviewer #1: (No Response)

Reviewer #2: Yes

4. Have the authors made all data underlying the findings in their manuscript fully available?

Reviewer #1: Yes

Reviewer #2: Yes

5. Is the manuscript presented in an intelligible fashion and written in standard English?

Reviewer #1: (No Response)

Reviewer #2: Yes

6. Review Comments to the Author

Reviewer #1: The authors addressed all of my concerns. Please correct the following few issues.

Line 34 delete the phrase "using IDEXX SNAP kits" from the abstract, not to make it appear endorsement of the kit.

A statement about E. coli, Ampicillin resistant E. coli and Tetracycline resistant E. coli results for the pasteurized milk may need to be mentioned in the abstract since pasteurized milk is also mentioned here.

Line 38: the numerator should be 74 (unpasteurized milk). Please make sure this has been corrected when results were reported as proportions of total samples analyzed (n=95 analyzed and not 96 collected) and when unpasteurized milk samples are considered separately (n=74 analyzed and not 75 collected)

Reviewer #2: This is a modest study that examines the presence of antibiotic residues in milk from vendors in Nairobi and seeks to determine if there are any correlations between presence of residues, and presence of resistant bacteria.There is a need to better understand antibiotic resistance in African countries, and this study provides information that informs question related to food safety and antibiotic resistance in a culturally relevant setting.

7. PLOS authors have the option to publish the peer review history of their article (what does this mean?). If published, this will include your full peer review and any attached files.

Reviewer #1: No

Reviewer #2: No

---

## [Editor Report · Acceptance letter]

13 May 2020

PONE-D-19-29752R1 

Antibiotic residues and antibiotic-resistant bacteria detected in milk marketed for human consumption in Kibera, Nairobi 

Dear Dr. Omulo:

I am pleased to inform you that your manuscript has been deemed suitable for publication in PLOS ONE. Congratulations! Your manuscript is now with our production department. 

With kind regards,

on behalf of

Dr. James E. Wells 

Academic Editor

PLOS ONE